# Disrupting tumour vasculature and recruitment of aPDL1-loaded platelets control tumour metastasis

Hongjun Li[1,2,3,4,5], Zejun Wang [3,4,5], Zhaowei Chen[1,3,6], Tianyuan Ci[3], Guojun Chen[3,4,5], Di Wen[3,4,5], Ruoxin Li[3,5], Jinqiang Wang[1,3,4,5], Huan Meng [5], R. Bryan Bell [7], Zhifeng Gu [8], Gianpietro Dotti [9] & Zhen Gu [1,2,3,4,5✉]

Although therapies of cancer are advancing, it remains challenging for therapeutics to reach the sites of metastasis, which accounts for majority of cancer associated death. In this study, we have developed a strategy that guides an anti-programmed cell death-ligand 1 (aPDL1) antibody to accumulate in metastatic lesions to promote anti-tumour immune responses. Briefly, we have developed a combination in which Vadimezan disrupts tumour blood vessels of tumour metastases and facilitates the recruitment and activation of adoptively transferred aPDL1-conjugated platelets. In situ activated platelets generate aPDL1-decorated platelet-derived microparticles (PMP) that diffuse within the tumour and elicit immune responses. The proposed combination increases 10-fold aPDL1 antibody accumulation in lung metastases as compared to the intravenous administration of the antibody and enhances the magnitude of immune responses leading to improved antitumour effects.

[1] College of Pharmaceutical Sciences, Zhejiang University, Hangzhou, Zhejiang, China. [2] Zhejiang University Medical Center, Sir Run Run Shaw Hospital, Hangzhou, Zhejiang, China. [3] Department of Bioengineering, University of California, Los Angeles, CA, USA. [4] Jonsson Comprehensive Cancer Center, University of California, Los Angeles, CA, USA. [5] California NanoSystems Institute, University of California, Los Angeles, CA, USA. [6] Institute of Food Safety and Environment Monitoring, College of Chemistry, Fuzhou University, Fuzhou, China. [7] Earle A. Chiles Research Institute in the Robert W. Franz Cancer Center, Providence Cancer Institute, Portland, OR, USA. [8] Research Center of Clinical Medicine, Affiliated Hospital of Nantong University, Nantong, China. [9] Joint Department of Biomedical Engineering, University of North Carolina at Chapel Hill and North Carolina State University, Raleigh, NC, USA. ✉email: guzhen@zju.edu.cn

Tumour metastasis remains challenging to eradicate and represents the principal cause of cancer-related mortality[1–3]. Once a tumour has metastasised, survival is poor, despite treatment with currently available therapies[4–6]. Recently, the clinical development of immune checkpoint blockade (ICB) targeting the PD-1/PD-L1 pathway has resulted in unprecedented improvements in overall survival within a broad range of malignancies, however, only a subgroup of patients experience durable benefit[7–10]. Challenges that hinder the treatment of metastasis are highly complex and related to an adaptable and heterogeneous biological environment, poor understanding of the molecular and cellular drivers of immune escape, alternations in the cytokine or chemokine milieu affecting cell trafficking to the tumour, tumour-associated organ dysfunction, tumour hypoxia, tumour volume, and drug resistance[11–16]. Furthermore, many systemic therapies are associated with toxicities and some can even be prometastatic[17,18]. Therefore, a strategy that enhances intratumoural delivery and simultaneously reduces the drug dosages to reach therapeutic effects represents an ideal approach to decreasing toxicity and increasing efficacy.

Angiogenesis is a crucial phenomenon causing tumour cell metastatic spreading[19,20], and multiple strategies have been developed to target the tumour neo-vasculature to treat cancer[19,21]. Anti-angiogenesis agents can cause local haemorrhage, activation of the coagulation cascade and platelet recruitment. The latter effect may be exploited to develop targeted drug delivery[22,23]. Herein, we report the combinatorial effects of a neo-blood vasculature disruption agent, Vadimezan, and aPDL1-conjugated platelets (P@aPDL1) to promote anti-tumour activity. Vadimezan is a small molecule inhibitor advanced to phase III clinical trials[22] that disrupts neo-blood vasculature and induces local tumour bleeding. We hypothesis that platelets loaded with aPDL1 and adoptively transferred can be preferentially recruited to the haemorrhagic tumour site, form platelet-derived microparticles[24,25] and locally release the aPDL1 to enhance T cell-based immunotherapy (Fig. 1a).

## Results

**Engineering platelets with aPDL1.** Mouse platelets were enriched from whole blood according to a standard protocol, and prostaglandin E1 (PEG1) was added to prevent platelet activation[26]. P@aPDL1 were prepared by covalently linking platelets and aPDL1 with a bio-functional linker, sulfosuccinimidyl-4-(N-maleimidomethyl)-cyclohexane-1-carboxylate (Sulfo-SMCC) (Supplementary Fig. 1). The successful conjugation was validated by confocal microscopy (CLSM) (Fig. 1b). Quantitively, enzyme-linked immunosorbent assay (ELISA) showed an aPDL1 conjugation rate of 0.3 pg/platelet. Surface proteins on the platelet involved in cell migration and adhesion were detected in P@aPDL1 and retained functions (Fig. 1c and Supplementary Fig. 2). The collagen-binding ability of P@aPDL1 was comparable to the native platelets (Fig. 1d). Upon platelet activation by thrombin stimulation, platelet-derived microparticles (PMPs) were generated and detected by transmission electron microscopy (TEM) (Fig. 1e). Approximately 70% of the surface-decorated aPDL1 was released from platelets by PMPs after eight hours of treatment (Supplementary Fig. 3).

**Vadimezan induces tumour vasculature disruption and platelet recruitment.** Vadimezan was reported to cause local bleeding in a murine colon carcinoma model[23,27]. In the 4T1 orthotopic breast tumour model, extensive haemorrhaging occurred 6 h after a single intravenous administration of 15 mg kg$^{-1}$ Vadimezan as compared to PBS treatment (Fig. 1f). The morphology of the tumour blood vessels was detected with immunofluorescence

staining. Distinguished from the homogeneous and complete tubular structures in tumour with PBS treatment, blood vessels became more fragmentised after Vadimezan administration (Fig. 1g) revealing the destruction of the tumour vessels. Notably, no significant haemorrhage occurred in the liver, kidney, and spleen (Supplementary Fig. 4) because Vadimezan specifically targets tumour endothelial cells and induces tumour necrosis factor-$\alpha$ (TNF-$\alpha$) secretion, which is known to cause haemorrhagic necrosis in tumours[22]. Vadimezan treatment caused recruitment of platelets within the haemorrhagic tumour and they were activated and identified in the tumour with CLSM (Fig. 1g).

We next evaluated whether treatment with Vadimezan can initiate micro-metastatic tumour haemorrhage and recruitment of transferred P@aPDL1. A metastatic tumour model was established by intravenous injection of the 4T1 cells into mice. Microtumours can be detected eight days post-injection (Supplementary Fig. 5). As shown in Fig. 2a, haemorrhagic deposits were observed in the tumours localised in the lung of mice with Vadimezan treatment, but not in those treated with PBS. Red blood cells (black arrow) observed in tumour sections analysed by H&E staining further indicated that Vadimezan induced bleeding in the micro-metastatic tumour.

To investigate whether Vadimezan treatment causes accumulation of aPDL1-conjugated platelets within the tumour, we monitored the accumulation in the lung and biodistribution of Cy5.5-labelled aPDL1. As shown in Fig. 2b, aPDL1 amount detected within the tumour was higher in the mice treated with P@aPDL1 and Vadimezan as compared to those treated with P@aPDL1 or free aPDL1 alone. Quantitative analysis showed that mice co-treated with Vadimezan and P@aPDL1 held 3- to 10-fold amount of aPDL1 in the tumour as compared to mice treated with P@aPDL1 and free aPDL1 antibody, respectively (Fig. 2c, d). These results demonstrated that combining P@aPDL1 with an antiangiogenic agent caused the accumulation of aPDL1 in the metastatic tumour, which could potentially augment the efficacy of ICB.

**P@aPDL1 combined with Vadimezan enhances anti-tumour effects in the breast metastatic tumour model.** We further investigated the anti-tumour ability of the P@aPDL1 and Vadimezan combination in 4T1-bearing mice. Ten days post-tumour inoculation, mice received one single administration of either PBS, free aPDL1, Vadimezan, P@aPDL1, Vadimezan combined with P@aPDL1 or Vadimezan combined with IgG-conjugated platelet (P@IgG) (Fig. 3a). The development of tumour metastasis was monitored by recording the bioluminescence of 4T1 cells in vivo using an in vivo imaging system (IVIS). As shown in Fig. 3b, c, mice that received P@aPDL1 combined with Vadimezan treatment showed significant tumour regression. Tumour bioluminescence signals in five out of eight mice were barely detectable 20 days after treatment. In contrast, mice treated with free aPDL1, Vadimezan, P@aPDL1 or Vadimezan combined with P@IgG showed no response, and most of the mice died within 40 days (Fig. 3d). Tumour-free survival at day 60 was 50% in mice receiving P@aPDL1 combined with Vadimezan treatment, and rare tumour lesions were observed in the lung at the time of the termination of the experiment (Fig. 3e–g). P@aPDL1 and Vadimezan combination treatment caused increased apoptosis of tumour cells as measured by terminal deoxynucleotidyl transferase dUTP nick end labelling (TUNEL) (Supplementary Fig. 6).

**P@aPDL1 combined with Vadimezan causes immune activation in the breast metastatic tumour model.** We investigated the immune response underlying the enhanced anti-tumour effects of the P@aPDL1 and Vadimezan combination therapy. T cells in

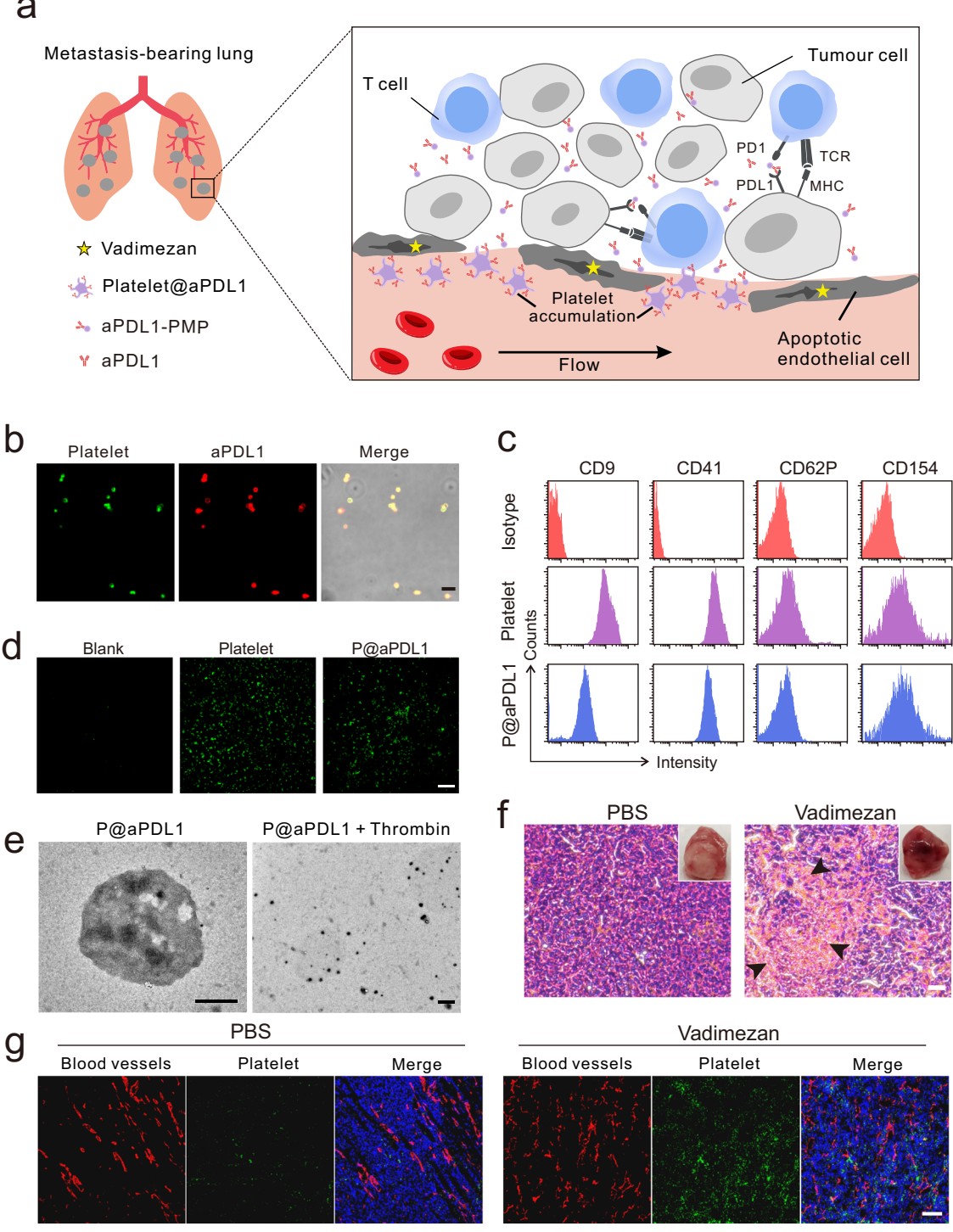

**Fig. 1 Characterisation of aPDL1-conjugated platelets and disruption of tumour blood vessel caused by Vadimezan. a** Schematic of using aPDL1-engineered platelets combined with Vadimezan to treat tumour metastases. **b** Confocal images of aPDL1-conjugated platelets (P@aPDL1). Platelets were labelled with WGA-488 (green) and the aPDL1 was tracked by Cy5-conjugated anti-rat IgG antibody (red), scale bar: 5 μm. **c** Flow cytometry analysis of the proteins on the native platelets and P@aPDL1. CD154 and CD62P were measured after the activation of the platelet. **d** Collagen-binding ability of native platelet and P@aPDL1. The blank was not pretreated with collagen, scale bar: 200 μm. **e** TEM image of the P@aPDL1 and platelet-derived microparticles (PMPs) generated by stimulation of P@aPDL1 with thrombin, scale bar: 500 nm. **f** The photograph and H&E staining image of the 4T1 tumour post-treatment with PBS and Vadimezan. Black arrows marked the haemorrhaging sites in the tumour, scale bar: 20 μm. **g** Confocal images of the blood vessels and activated platelets in the 4T1 tumours treated with Vadimezan. Blood vessels were marked with an anti-CD31 antibody (red), the activated platelets were labelled with anti-CD62P antibody (green), and the nucleus were stained with Hoechst 33342 (blue), scale bar: 100 μm.

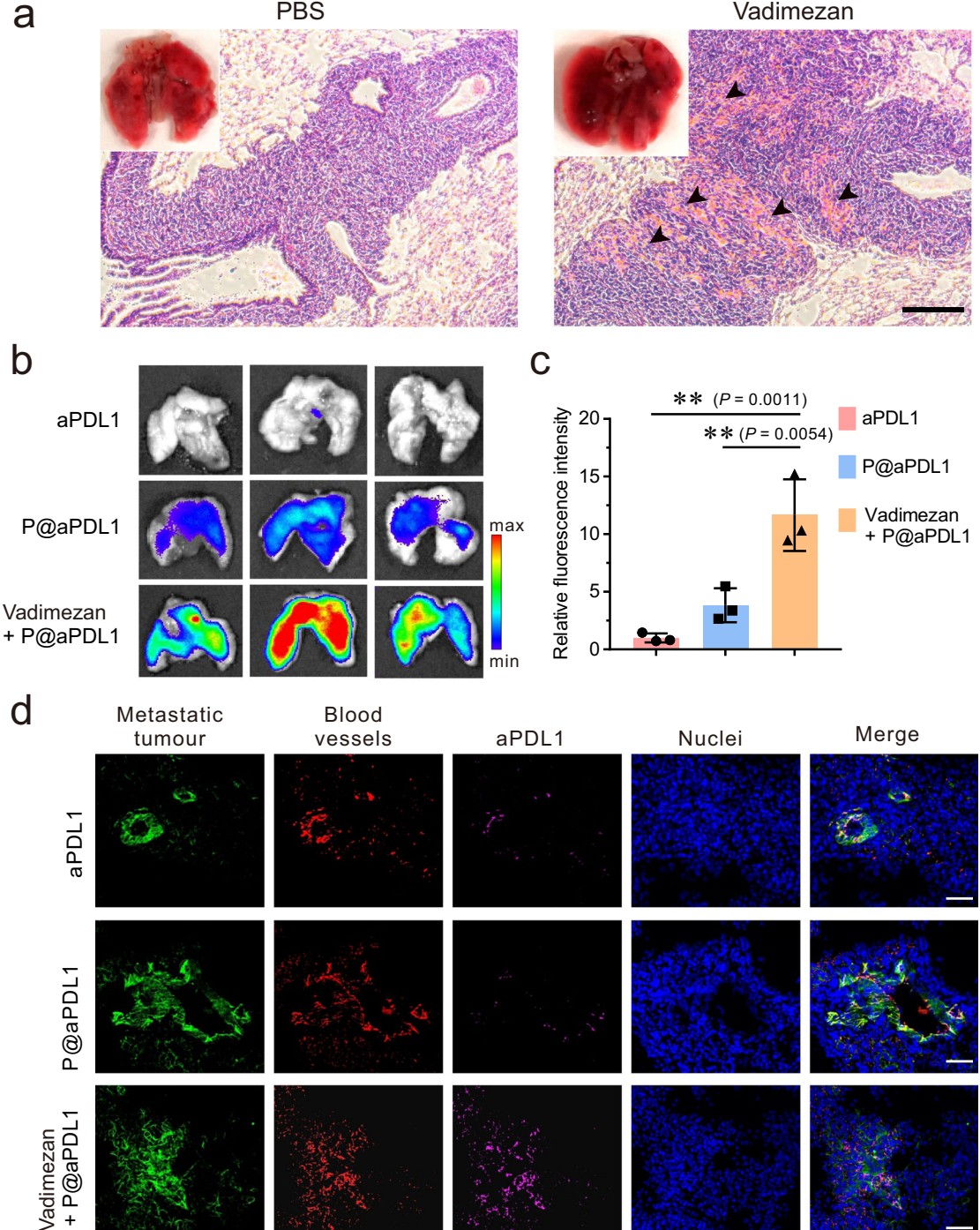

**Fig. 2 P@aPDL1 recruitment at the tumour site after Vadimezan treatment. a** Photographs and H&E staining images of the 4T1 tumour metastasis in the lung post-treatment with PBS and Vadimezan. Black arrows marked the haemorrhaging sites within the tumour, scale bar: 100 μm. **b** Fluorescence images showing the accumulation of aPDL1-Cy5.5 in the tumour metastases within the lung. **c** Quantitative analysis of the fluorescence intensity of aPDL1 in the lungs, $n = 3$ biologically independent animals, data are prestented as mean ± SD. **d** Confocal images showing the aPDL1 distribution within metastases in the lung. Metastatic tumour cells expressed GFP (green), blood vessels were marked by AF568-anti-CD31 antibody (red), and the aPDL1 was marked by Cy5-conjugated anti-rat IgG antibody (magenta), nuclei were labelled by Hoechst 33342 (blue), scale bar: 50 μm. Data are presented as mean ± SD, statistical significance was analysed via ANOVA (one-way, Tukey post-hoc test). $P$-value: ** $P < 0.01$.

peripheral blood (PB) were analysed seven days post-treatment (Supplementary Fig. 7). We detected a 1.5-fold increase of CD3+ T lymphocytes in mice with P@aPDL1 and Vadimezan combination treatment as compared with other treatment groups (Fig. 4a and Supplementary Fig. 8). Furthermore, mice that received P@aPDL1 and Vadimezan combination treatment

displayed a 1.3-fold increase in CD4+ T cells and a 2.2-fold increase in CD8+ T cells as compared to control groups (Fig. 4b–d). Immunofluorescence staining also revealed more CD4+ and CD8+ T cells in metastatic lung tumours in mice with P@aPDL1 and Vadimezan treatment as compared to control groups (Fig. 4e and Supplementary Fig. 9). Circulating cytotoxic

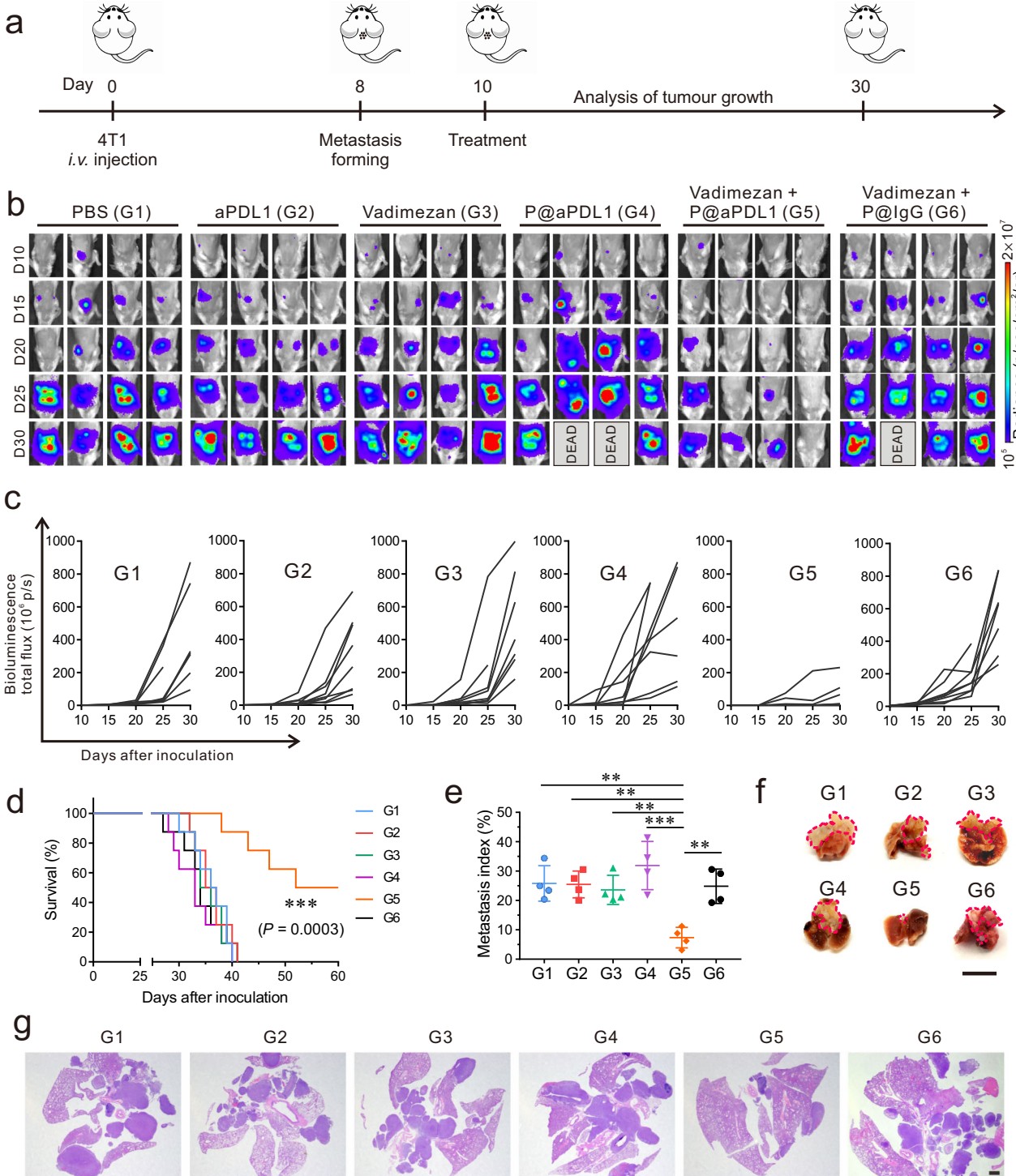

**Fig. 3 P@aPDL1 combined with Vadimezan promotes anti-tumour effects in the 4T1 tumour model. a** Schematics of the 4T1 breast cancer metastatic model and treatment. Mice received were treated 10 days post-tumour cells inoculation. Mice received 1 mg kg$^{-1}$ aPDL1 antibody or 15 mg kg$^{-1}$ Vadimezan. **b** In vivo tumour bioluminescence images of mice that received different treatments. **c** Tumour bioluminescence intensity growth kinetics in different groups. **d** Survival rates of mice treated in different groups, $n = 8$ biologically independent animals. **e** Metastatic index (tumour region/total region) and **f** representative lung photograph and **g** H&E staining of tumour lesions after different treatments. $n = 4$ biologically independent samples in **e**, pink line indicate the tumours in the lung parenchyma in **f**. Scale bar for **f** is 1 cm, scale bar for **g** is 1 mm. G1, PBS; G2, aPDL1; G3, Vadimezan; G4, P@aPDL1; G5, P@aPDL1 + Vadimezan; G6, P@IgG + Vadimezan. Data are presented as mean ± SD, survival statistical significance was analysed by log-rank (Mantel–Cox) test, and the statistical significance in **e** was analysed via ANOVA (one-way, Tukey post-hoc test). P-value: **$P < 0.01$, ***$P < 0.001$. **e** G1 vs. G5: $P = 0.0011$, G2 vs. G5: $P = 0.0013$, G3 vs. G5: $P = 0.0034$, G4 vs. G5: $P < 0.0001$, G6 vs. G5: $P = 0.0018$.

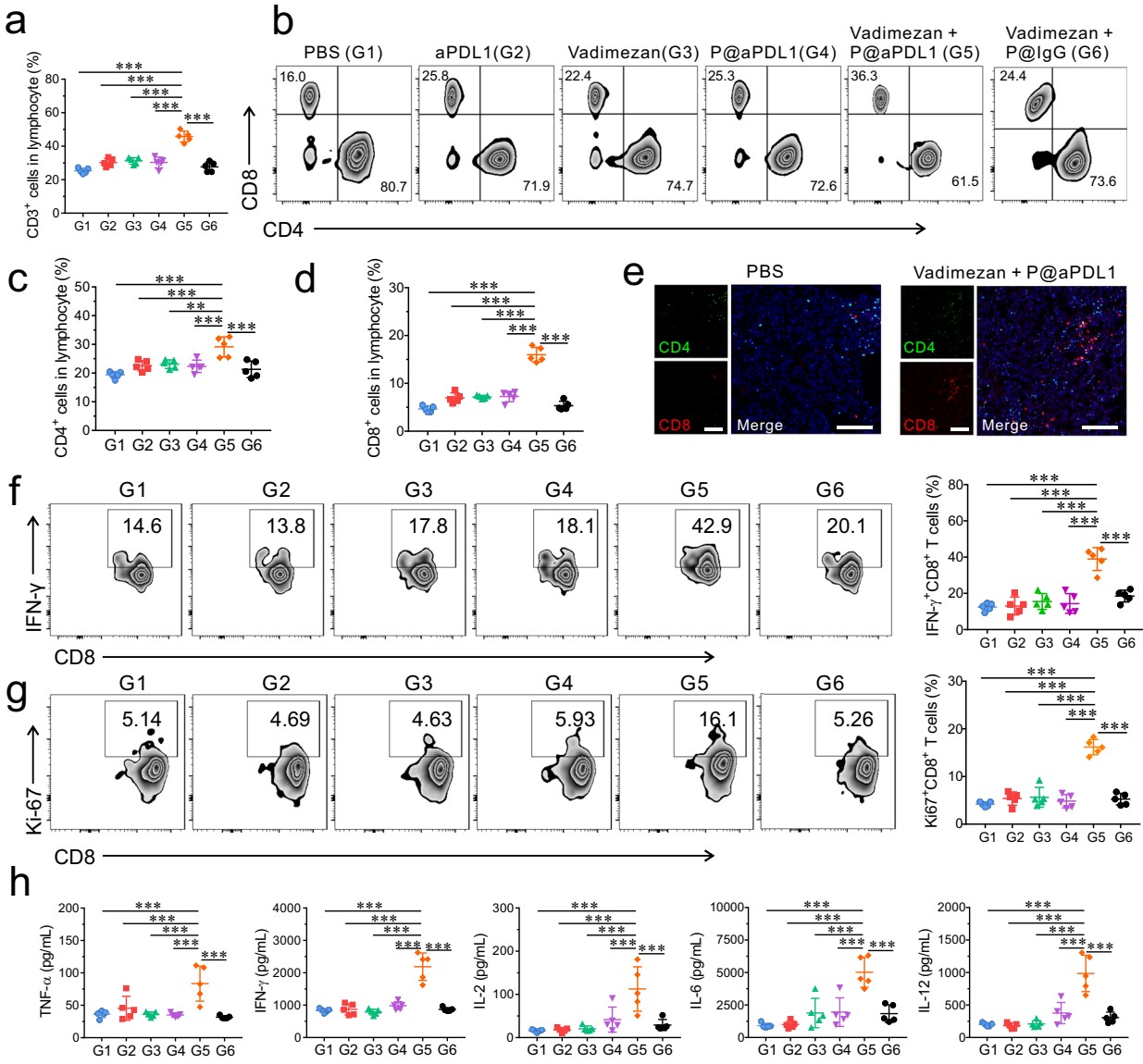

**Fig. 4 Immune responses post P@aPDL1 and Vadimezan treatment in the 4T1 tumour model. a** Relative quantification of the CD3$^+$ T cells in the PB, $n =$ 5 biologically independent animals. **b** Representative flow cytometry analysis of CD4$^+$ T cells, CD8$^+$ T cells gating on CD3$^+$ T cells. Relative quantification of the **c** CD4$^+$ T cells and **d** CD8$^+$ T cells, $n = 5$ biologically independent animals. **e** Representative immunofluorescence images of metastatic tumour in the lung showing CD4$^+$ T cells (green) and CD8$^+$ T cells (red) after treatment, scale bar: 100 μm. Representative flow cytometry analysis of **f** IFN-$\gamma^+$CD8$^+$ T cells and **g** Ki-67$^+$CD8$^+$ T cells gating on CD3$^+$ T cells, $n = 5$ biologically independent animals. **h** Cytokine levels in the serum of mice, $n = 5$ biologically independent animals. G1, PBS; G2, aPDL1; G3, Vadimezan; G4, P@aPDL1; G5, P@aPDL1 + Vadimezan; G6, P@IgG + Vadimezan. Data are presented as mean ± SD, statistical significance was analysed via ANOVA (one-way, Tukey post-hoc test). *P*-value: **$P < 0.01$, ***$P < 0.001$. **a** G1 vs. G5: $P < 0.0001$, G2 vs. G5: $P < 0.0001$, G3 vs. G5: $P < 0.0001$, G4 vs. G5: $P < 0.0001$, G6 vs. G5: $P < 0.0001$. **c** G1 vs. G5: $P < 0.0001$, G2 vs. G5: $P = 0.0007$, G3 vs. G5: $P =$ 0.0018, G4 vs. G5: $P = 0.0005$, G6 vs. G5: $P < 0.0001$. **d** G1 vs. G5: $P < 0.0001$, G2 vs. G5: $P < 0.0001$, G3 vs. G5: $P < 0.0001$, G4 vs. G5: $P < 0.0001$, G6 vs. G5: $P < 0.0001$. **f** G1 vs. G5: $P < 0.0001$, G2 vs. G5: $P < 0.0001$, G3 vs. G5: $P < 0.0001$, G4 vs. G5: $P < 0.0001$, G6 vs. G5: $P < 0.0001$. **g** G1 vs. G5: $P <$ 0.0001, G2 vs. G5: $P < 0.0001$, G3 vs. G5: $P < 0.0001$, G4 vs. G5: $P < 0.0001$, G6 vs. G5: $P < 0.0001$. **h** TNF-α: G1 vs. G5: $P < 0.0001$, G2 vs. G5: $P =$ 0.0009, G3 vs. G5: $P < 0.0001$, G4 vs. G5: $P < 0.0001$, G6 vs. G5: $P < 0.0001$; IFN-γ: G1 vs. G5: $P < 0.0001$, G2 vs. G5: $P < 0.0001$, G3 vs. G5: $P < 0.0001$, G4 vs. G5: $P < 0.0001$, G6 vs. G5: $P < 0.0001$; IL-2: G1 vs. G5: $P < 0.0001$, G2 vs. G5: $P < 0.0001$, G3 vs. G5: $P < 0.0001$, G4 vs. G5: $P = 0.0007$, G6 vs. G5: $P <$ 0.0001; IL-6: G1 vs. G5: $P < 0.0001$, G2 vs. G5: $P < 0.0001$, G3 vs. G5: $P < 0.0001$, G4 vs. G5: $P < 0.0001$, G6 vs. G5: $P < 0.0001$; IL-12: G1 vs. G5: $P < 0.0001$, G2 vs. G5: $P < 0.0001$, G3 vs. G5: $P < 0.0001$, G4 vs. G5: $P < 0.0001$, G6 vs. G5: $P < 0.0001$.

T lymphocytes (IFN-$\gamma^+$CD8$^+$ T cells) were 2.7~3.1-fold higher in mice receiving P@aPDL1 and Vadimezan treatment compared to control mice (Fig. 4f). Similarly, a higher number of proliferating T cells (Ki67$^+$CD8$^+$ T cells) were detected in mice administrated with P@aPDL1 and Vadimezan (Fig. 4g). More cytotoxic T lymphocytes (IFN-$\gamma^+$CD8$^+$ T cells) were detected in the tumour, supporting the elicitation of immune responses within the

tumour site (Supplementary Fig. 10). Cytokines including TNF-α, IFN-γ, IL-2, IL-6, and IL-12 were also increased in the PB (Fig. 4h) and in tumour lesions (Supplementary Fig. 11) of mice with P@aPDL1 and Vadimezan treatment. No increase of immune response, including numbers of CD3$^+$, CD4$^+$ and CD8$^+$ T cells, was detected in the non-tumour-bearing mice treated with P@aPDL1 and Vadimezan compared with those treated with PBS

(Supplementary Fig. 12), which further indicates that the tumour specificity of the observed enhanced immune response in the metastatic breast tumour model.

**P@aPDL1 combined with Vadimezan causes anti-tumour effects in the B16 tumour model.** To examine the broad applicability of the combined P@aPDL1 and Vadimezan treatment we used the B16 melanoma metastatic model. B16F10 cells were intravenously inoculated into C57BL/6 mice and mice were subsequently treated with P@aPDL1 and Vadimezan and compared to control treatments (Fig. 5a). Tumour metastasised rapidly in mice treated with aPDL1, Vadimezan, and P@aPDL1, similar to the trend of the PBS treatment group (Fig. 5b). In contrast, the development of melanoma metastasis was considerably reduced in mice with P@aPDL1 and Vadimezan treatment, and one out of five mice (20%) showed tumour progression within 26 days post-B16F10 inoculation (Fig. 5b, c). Quantitative analysis of the tumour bioluminescence signal further substantiated the enhanced anti-tumour effects of the P@aPDL1 and Vadimezan treatment (Fig. 5c). Forty per cent of mice with P@aPDL1 and Vadimezan treatment survived at least 60 days as compared to control groups (Fig. 5d), and fewer lung metastases were observed after euthanasia (Fig. 5e). As observed in the 4T1 tumour model, treatment with P@aPDL1 and Vadimezan caused approximately a 1.5-fold increase of $CD3^+$ T cells in the PB (Fig. 5f and Supplementary Figs. 13, 14), increase in $CD8^+$ T cells, central memory $CD8^+$ T cells ($CD8^+CD62L^+CD44^+$), effector memory CD8 T cells ($CD8^+CD62L^-CD44^+$) and naïve CD8 T cells ($CD8^+CD62L^+CD44^-$) (Fig. 5g–j). We also observed more cytotoxic T lymphocytes (IFN-$\gamma^+CD8^+$ T cells) in the tumour after the P@aPDL1 and Vadimezan combined treatment compared to other control groups (Supplementary Fig. 15). To further validate the broad applicability of the combined therapy, we treated the mice bearing the metastatic Lewis lung cancer with the P@aPDL1 and Vadimezan. The combined treatment with P@aPDL1 and Vadimezan significantly prolonged the survival of the mice (Supplementary Fig. 16).

## Discussion

To date, a variety of strategies have been advanced to initiate tumour-specific T cell responses in patients, such as ICB, engineered T cells, interferons/interleukins, and tumour vaccines. While these immunotherapeutic approaches are transforming the practice of oncology across a broad spectrum of both solid and hematologic malignancies, response rates are limited, particularly in tumours that lack T cells or an inflamed phenotype. Targeted delivery of the immunotherapeutic agents directly to the metastatic tumours could potentially enhance treatment efficacy as well as reduce side effects[28–30]. By inducing haemorrhagic necrosis at the tumour site coupled with engineered platelet recruitment, this approach leverages the physiological interactions of platelet and tumour vasculature to augment targeting capability.

Recent studies show that platelets promote tumour metastasis by physical association and bidirectional activation. Specifically, cloaking protects circulating tumour cells from shear forces and attack by the immune system, while promoting vascular adhesion of tumour cells and its transendothelial migration[31,32]. In our study, we converted platelets into a "Trojan horse" for tumour targeting and loaded with ICB for enhanced tumour killing. Importantly, our proposed strategy aims to develop a method for treatment of already formed metastatic tumours which frequently cause the patient's death. Moreover, the current therapeutic achievement is based on a one-time treatment, therefore,

administration frequency adjustment could further potentiate the efficacy against metastasis.

Recent studies demonstrated that the stimulator of the interferon gene (STING) is the receptor of Vadimezan, which activates the TBK1/IRF3 (interferon regulatory factor 3) and NF-$\kappa$B (nuclear factor $\kappa$B) signalling pathways to induce robust type I interferon and proinflammatory cytokine responses[33,34]. Consistent with the fact that STING activation upregulates PD-L1 expression in tumour cells[35], we observed PD-L1 upregulation in 4T1 breast tumours following Vadimezan treatment (Supplementary Fig. 17), providing the rationale for combining Vadimezan with ICB. While Vadimezan did not show synergistic effects with chemotherapy in phase III clinical trials, our proposed therapy offers a combination that links the biological properties of the Vadimezan with the physiologic functions of platelets to create an effective combined immunotherapy strategy. Future studies will assess alternative combinatorial strategies, dose, and timing of the combination of P@PD-L1 and Vadimezan we propose. Importantly, the long-term storage of engineered platelets can be further adapted to develop a "off-the-shelf" cell therapy for metastatic tumours.

In summary, we have developed a strategy that combines a tumour vasculature disrupting agent (Vadimezan) with aPDL1-loaded platelets to target tumour metastases. Vadimezan induces tumour vascular endothelial cell apoptosis, which leads to haemorrhagic necrosis at the tumour site. This milieu promotes the trafficking of aPDL1-engineered platelets and their activation to deliver payload aPDL1-PMPs within the tumour site that elicit immune responses. This approach leverages the dynamic role of platelets in human physiology and represents a promising treatment for metastatic cancer.

## Methods

**Cell lines and mice.** Mouse mammary carcinoma cell line 4T1 and mouse melanoma cell line B16F10 and were purchased from the American Type Culture Collection. The B16F10-Luc cell line was purchased from Imanis Life Sciences, the 4T1-Luc cell line and the 4T1-GFP cell line were a gift from Prof. H. Meng at the University of California, Los Angeles (UCLA). Cells were incubated in DMEM (Gibco, Invitrogen) with 10% FBS (Invitrogen) and 100 U mL$^{-1}$ penicillin/streptomycin (Invitrogen) at 37 °C in 5% $CO_2$.

C57BL/6 and BALB/c mice were purchased from The Jackson Laboratory. Age-matched (6–10 weeks) female mice were used throughout all experiments. All mouse studies were carried out under the protocols approved by the Institutional Animal Care and Use Committee at the University of California, Los Angeles (UCLA).

**Antibodies.** The anti-mouse PD-L1 antibody was purchased from BioLegend Inc. (catalogue no. 124329, clone: 10F.9G2). The antibodies used for flow cytometry analysis and immunofluorescence staining including: CD3 (BioLegend, catalogue no. 100336, clone: 145-2C11, Brilliant Violet 421-conjugated), CD4 (BioLegend, catalogue no. 100408, clone: GK1.5, PE-conjugated), CD4 (BioLegend, catalogue no. 100406, clone: GK1.5, FITC-conjugated), CD8 (Invitrogen, catalogue no. 2023410, clone: 53-6.7, APC-conjugated), Ki67 (BioLegend, catalogue no. 652410, clone: 16A8, FITC-conjugated), IFN-$\gamma$ (BioLegend, catalogue no. 505806, clone: XMG1.2, FITC-conjugated), CD44 (BioLegend, catalogue no. 103024, clone: 1M7, PE-conjugated), CD62L (BioLegend, catalogue no. 304823, clone: DREG-56, PerCP/Cyanine5.5-conjugated), CD9 (BioLegend, catalogue no. 124807, clone: MZ3, FITC-conjugated), CD41 (BioLegend, catalogue no. 133904, clone: MWReg30, PE-conjugated), CD36 (BioLegend, catalogue no. 102605, clone: HM36, PE-conjugated), CD61 (BioLegend, catalogue no. 104307, clone: 2C9.G2, PE-conjugated), CD62P (BioLegend, catalogue no. 148305, clone: RMP-1, PE-conjugated), CD154 (BioLegend, catalogue no. 106505, clone: MR1, PE-conjugated), PD-L1 (BioLegend, catalogue no. 124308, clone: 10F.9G2, PE-conjugated), CD62P (R&D Systems, catalogue no. AF737, goat IgG), CD31 (R&D Systems, catalogue no. AF3628, goat IgG), Donkey anti-goat IgG (Invitrogen, catalogue no. 2044862, AF568-conjugated), goat anti-rat IgG (Invitrogen, catalogue no. 2040140, APC-conjugated). The antibodies were diluted 50-200 times during staining depends on its brightness.

**Preparation of aPDL1-conjugated platelets.** Platelets were obtained from the blood of mice. Typically, the mouse blood was collected into dipotassium ethylenediaminetetraacetic acid-treated tubes, and the supernatant was obtained with

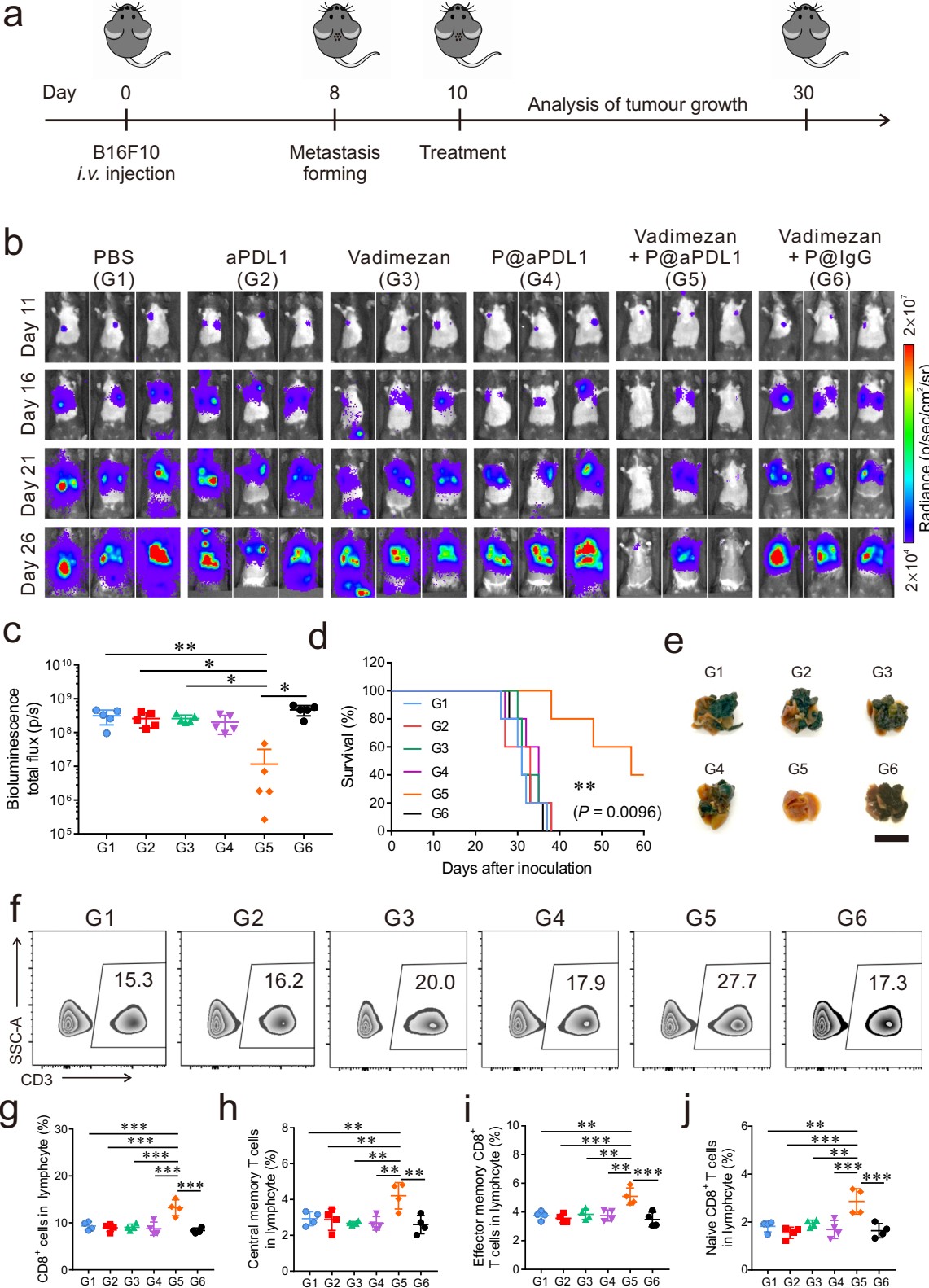

centrifugation of the blood (100 × g, 8 min, 22 ℃). Then Prostaglandin E1 (PGE1, Sigma) was added into the supernatant with a concentration of 1 μM, the supernatant was further centrifuged at 1000 × g for ten minutes to collect the platelets. PBS containing 1 μM PGE1 was used to resuspend the platelets. The platelets number was calculated using a hemocytometer under a microscope. The isolated platelets (1 × 10⁸) were then incubated with Traut's reagent (2-Iminothiolane, Sigma) (0.1 mg mL⁻¹) for two hours at 22 ℃ to convert the amine group to thiol group on the surface of platelets. The excess Traut's Reagent was removed under centrifugation (8 min, 1000 × g), and further resuspended with PBS with PGE1. In the meantime, the aPDL1 was mixed with sulfosuccinimidyl-4-(N-maleimidomethyl) cyclohexane-1-carboxylate (Sulfo-SMCC, Thermo Scientific) (1/1.2, molar ratio, aPDL1/Sulfo-SMCC) for 2 h at 4 ℃. The unreacted Sulfo-SMCC was removed by centrifugation with a centrifugal filter tube (molecular weight cut-off =10 kDa). Lastly, antibodies were mixed with platelets in PBS containing PGE1

**Fig. 5 P@aPDL1 combined with Vadimezan promotes anti-tumour effects in the B16 tumour model. a** Schematics of the melanoma metastatic model and treatment. Mice were treated 10 days post-tumour cells inoculation. **b** In vivo tumour bioluminescence images of mice receiving different treatments. **c** Quantitative analysis of the tumour bioluminescence intensity at day 26, $n = 5$ biologically independent animals. **d** Survival rate of the mice in different groups, $n = 5$ biologically independent animals. **e** Representative lung photographs after treatment, scale bar: 1 cm. **f** Representative flow cytometry analysis of CD3$^+$ T cells in the PB. The relative quantification of the **g** CD8$^+$ T cells, **h** central memory CD8$^+$ T cells (CD8$^+$CD62L$^+$CD44$^+$), **i** effector memory CD8 T cells (CD8$^+$CD62L$^-$CD44$^+$) and **j** naive CD8 T cells (CD8$^+$CD62L$^+$CD44$^-$) in the blood 14 days after treatment, $n = 4$ biologically independent animals. G1, PBS; G2, aPDL1; G3, Vadimezan; G4, P@aPDL1; G5, P@aPDL1 + Vadimezan; G6, P@IgG + Vadimezan. All the data are presented as mean ± SD, the statistical significance of the survival was calculated via log-rank (Mantel–Cox) test, and others were calculated via one-way ANOVA with a Tukey post-hoc test. P-value: \*$P < 0.05$, \*\*$P < 0.01$, \*\*\*$P < 0.001$. **c** G1 vs. G5: $P = 0.0017$, G2 vs. G5: $P = 0.0101$, G3 vs. G5: $P = 0.0108$, G4 vs. G5: $P = 0.0572$, G6 vs. G5: $P < 0.0001$. **g** G1 vs. G5: $P = 0.0004$, G2 vs. G5: $P < 0.0001$, G3 vs. G5: $P = 0.0002$, G4 vs. G5: $P < 0.0001$, G6 vs. G5: $P < 0.0001$. **h** G1 vs. G5: $P = 0.0076$, G2 vs. G5: $P = 0.0055$, G3 vs. G5: $P = 0.0016$, G4 vs. G5: $P = 0.0016$, G6 vs. G5: $P = 0.001$. **i** G1 vs. G5: $P = 0.0014$, G2 vs. G5: $P = 0.0004$, G3 vs. G5: $P = 0.0025$, G4 vs. G5: $P = 0.0014$, G6 vs. G5: $P = 0.0002$. **j** G1 vs. G5: $P = 0.0012$, G2 vs. G5: $P < 0.0001$, G3 vs. G5: $P = 0.0034$, G4 vs. G5: $P = 0.0004$, G6 vs. G5: $P = 0.0002$.

(1 μM). The aPDL1-conjugated platelets were collected with centrifugation (8 min, 1000 × $g$, 4 h). The amount of aPDL1-conjugated to the platelet was determined by ELISA (eBioscience, catalogue no. 88-50490-22).

**Characterisation of P@aPDL1.** For the collagen-binding assay, the collagen solution (mouse collagen type I/III, 2.0 mg mL$^{-1}$, Bio-Rad) was added to a glass-bottom 96-well plate and incubated overnight at 4 °C. The native platelets or the P@aPDL1 were resuspended with BSA solution (2%) to block nonspecific interaction (30 min). Then the platelets or P@aPDL1 were stained with Wheat Germ Agglutinin (WGA, Alexa Fluor™ 488 Conjugate, Invitrogen) for 30 min and washed with PBS. The platelets were added to the collagen-coated and non-coated plate (~1 × 10$^7$ cells/well) with 30 s incubation. The platelets binding to the plate were detected under fluorescence microscopy (Nikon) after washed with PBS.

For the imaging analysis of P@aPDL1, the P@aPDL1 were stained with Wheat Germ Agglutinin (WGA, Alexa Fluor™ 488 Conjugate, Invitrogen) and a goat anti-rat IgG (APC-conjugated) for 30 min. After washed by PBS, the P@aPDL1 was observed with a confocal microscope (Zeiss LSM880).

**Transmission electron microscopy.** P@aPDL1 and thrombin treated P@aPDL1 were fixed in paraformaldehyde solution (4%) for 30 min. Afterwards, the P@aPDL1 was collected by centrifugation (8 min, 1000 × $g$, 22 °C). For thrombin treated P@aPDL1 (PMP) collection, the thrombin treated P@aPDL1 was first centrifuged, then collected supernatant and centrifuged at 15,000 × $g$ for 30 min; then PMP was collected, incubated with 2% uranyl acetate, then mixed with lead citrate for 5 min, and then transferred to a copper grid. The TEM images were obtained with a T12 cryo-electron microscope (FEI Tecnai).

**aPDL1 release from P@aPDL1.** Briefly, 1 × 10$^8$ P@aPDL1 was suspended in PBS (1 mL), and thrombin with a final concentration of 0.5 U mL$^{-1}$ was used for stimulation of the platelets. At a different time, 50 μL samples were extracted and further centrifuged for 8 min at 1000 × $g$. The supernatant containing aPDL1 was collected and the amount of aPDL1 was measured by ELISA kit.

**In vivo tumour models and treatment.** For a study of tumour vasculature disruption after treatment with Vadimezan, an orthotopic breast tumour was established by transplanting 4T1 cells (1 × 10$^6$) to the mammary gland fat pad. PBS or Vadimezan (15 mg kg$^{-1}$, Tocris Bioscience) was intravenously injected when the tumours reached 200 mm$^3$. Six hours post-injection, the mice were sacrificed, the tumours were collected to analyse the haemorrhage and platelet recruitment by immunofluorescence staining.

For a study of treatment efficacy in the metastasis model, luciferase or GFP-tagged 4T1 cells (5 × 10$^5$) were injected into the BALB/c mice intravenously. The formation and development of lung metastases were tracked by monitoring the bioluminescence signals with the IVIS, (PerkinElmer, Live image 4.0). Eight days later, mice were randomly separated into five groups ($n = 8$). The mice treated with different formulations, including PBS, aPDL1, Vadimezan, P@aPDL1, and Vadimezan + P@aPDL1 (aPDL1: 1 mg kg$^{-1}$, Vadimezan: 15 mg kg$^{-1}$). The bioluminescence signals were recorded on day 10, 15, 20, 25 and 30 after tumour cells inoculation. The Living Image Software was used for the analysis of signals.

Another mouse melanoma metastatic tumour model was generated by injecting B16F10 cells into the C57BL6 mice intravenously. Briefly, mice were intravenously injected with 5 × 10$^5$ luciferase-tagged B16F10 cells, and 10 days post-cancer cell inoculation, mice were randomly separated into five groups ($n = 5$). The mice received treatments with different formulations, including PBS, aPDL1, Vadimezan, P@aPDL1, and Vadimezan + P@aPDL1 (aPDL1: 1 mg kg$^{-1}$, Vadimezan: 15 mg kg$^{-1}$). The bioluminescence signals were recorded on day 11, 16, 21 and 26 after tumour cells inoculation.

For observation of tumour growth in the lungs, mice that received different treatments were sacrificed at day 30 (for 4T1 metastasis model) or day 25 (for B16F10 melanoma metastasis model) post-cancer cell inoculation, and the tumour-bearing lung tissues were fixed with paraformaldehyde (4%) solution and stained with H&E to observe the metastasis on the lungs.

**aPDL1 accumulation in metastasis-bearing lung tissues.** To study aPDL1 accumulation in metastasis-bearing lung tissues, aPDL1 was labelled by Cyanine5.5 NHS-ester (Lumiprobe) before conjugated to the platelets. After purification, the Cy5.5-labelled aPDL1, P@aPDL1, and Vadimezan + P@aPDL1 were administrated to the mice intravenously (10 days after 4T1 cells inoculation). Twenty-four hours post-administration, lungs were collected, then the aPDL1 in the lungs were detected by IVIS, the fluorescence signals were analysed by Living Image Software.

**Flow cytometry.** Blood samples were harvested (breast metastasis, 7 days; melanoma metastasis, 14 days) post-treatment. The cells were stained with antibodies post lysis process to remove the red blood cells, all of the antibodies stainings was followed the manufacturer's instructions. The stained cells were measured on a BD LSR II flow cytometer with BD FACSDiva 6.1 and analysed with FlowJo 10.7 software.

**Cytokine measurement.** The plasma and metastatic site levels of cytokines, including TNF-α (BioLegend, catalogue no. 430904), IFN-γ (BioLegend, catalogue no. 430801), IL-2 (BioLegend, catalogue no. 431001), IL-6 (BioLegend, catalogue no. 430604), and IL-12 (BioLegend, catalogue no. 430304) were measured with ELISA 7-day post-treatment. The plasma samples were obtained with centrifugation (2000 × $g$, 4 °C), and the lung tissue samples were harvested by homogenised in cold PBS buffer in the presence of digestive enzymes and further centrifugated at 2000 × $g$ at 4 °C.

**Immunofluorescence staining.** Tumours were harvested from the mice and frozen in the cutting medium before sectioning via a cryotome. Before staining, the tissues were incubated with 1% BSA to block the nonspecific binding. For blood vessel staining, 10 μg mL$^{-1}$ of anti-mouse CD31 (R&D Systems, catalogue no. AF3628, goat IgG) antibody was incubated with the slides overnight, and the donkey anti-goat IgG (Invitrogen, catalogue no. 2044862, AF568-conjugated) secondary antibody was added after three times PBS washing, the nuclei were stained with Hoechst 33342. For platelet staining, 10 μg mL$^{-1}$ of anti-mouse CD62P (R&D Systems, catalogue no. AF737, goat IgG) were used. For the T cells staining, 20 μg mL$^{-1}$ PE anti-mouse CD8a antibody (Biolegend, cat no. 100708, clone: 53-6.7) and Alexa Fluor® 488 anti-mouse CD4 antibody (Biolegend, cat no. 100425, clone: GK1.5) was used to stain the tumour slices overnight at 4 °C. The tumour cell apoptosis was analysed with TUNEL. The stained slides were measured on a Zeiss LSM880 confocal laser scanning microscope and analysed with Zen 2 and Image J software.

**Statistics and reproducibility.** All results are presented as the mean ± standard deviation (SD). Tukey post-hoc tests and one-way ANOVA were used for multiple comparisons, and Student's $t$-test was used for two-group comparisons. The survival benefit was determined using a log-rank (Mantel–Cox) test. All statistical analyses were calculated by Prism (PRISM 7.0). The threshold for statistical significance was $P < 0.05$. All the experiments were repeated independently with similar results at least three times.

**Reporting summary.** Further information on research design is available in the Nature Research Reporting Summary linked to this article.

## Data availability

All the other data supporting the findings of this study are available within the article and its supplementary information files and from the corresponding author upon reasonable request. Source data for the figures are available in Figshare at https://doi.org/10.6084/m9.figshare.14099444.v1.

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

## Acknowledgements

We acknowledge Broad Stem Cell Reseach Center (BSCRC) Core Facilities, and Crump Institute-Preclinical Imaging Technology Center at UCLA for providing the analytical Instruments. This work was supported by grants from the start-up packages of Zhejiang University, start-up packages of UCLA, National Institutes of Health (R01 CA234343-01A1), and Jonsson Comprehensive Cancer Center at UCLA.

## Author contributions

Z.G. and H.L. were responsible for the conception and experimental strategy of the study. H.L, Z.W., T.C., Z.C., G.C., D.W., R.L. and J.W. performed the experiments and acquired the data. H.L, Z.W., Z.G. and Z.G. interpreted the data. H.L., Z.W., H.M, R.B.B, G.D, Z.G. and Z.G. co-wrote the manuscript.

## Competing interests

Z.G. is the co-founder of Zcapsule Inc. The other authors declare no competing interests.
