## [Peer Review File · Nature Communications]

REVIEWER COMMENTS

Reviewer #1 (Remarks to the Author):

This well written manuscript describes murine testing of a novel therapy in which anti-PD-L1 is conjugated to platelets. These cell conjugates can deliver more PD-L1 to tumor compared to standard soluble administration when administered with anti-angiogenic compound Vadimezan. The therapeutic was tested in two anti-PD-L1 resistant animal models and demonstrates improved tumor control relative to either the cell conjugate or Vadimezan alone. While the manuscript is well written grammatically, there are a few key issues with the manuscript that greatly weaken it. Importantly, the immunology is very weak overall. There is no evidence that the immune responses observed following administration of the treatment are specific for the tumor (i.e., no antigen-specificity). Some of the immunologic data do not make sense. For example, the authors claim that a single treatment results in nearly 40% of the circulating CD8 T cells as becoming activated by IFN gamma production. There are no other assay results that support this finding. In another example, in Figure 5F, the authors claim that CD3 T cells increase in the peripheral blood by ~50 with the combination. However, the scattergrams are not typical showing a homogeneous group of cells by side scatter analysis. This is not usual pattern seen in blood of mice. Furthermore, the changes are reported as fractional changes which may suggest that there was either a lymphoproliferation or there was a decrease in some other population. Thus, the nature of these changes is unclear. It is also unclear if any of the changes observed are linked to tumor as the experimental design lacked important controls including platelets conjugated to an irrelevant antibody and testing of the compound in non-tumor bearing animals. On a minor note, The IFN and TNF graphs in S10 look identical and should be checked.

Reviewer #2 (Remarks to the Author):

In this manuscript, the authors developed a strategy that guided an anti-programmed cell death-ligand 1 (aPDL1) antibody to efficiently accumulate in metastatic lesions to promote anti-tumor immune responses. Briefly, by giving a vasculature disruption agent to induce local hemorrhage, activation of the coagulation cascade in advance, the subsequent administered aPDL1-conjugated platelets (P@aPDL1) can significantly target to metastasis sites. Overall, this is an interesting and novel piece of work and the major conclusions are fully supported and demonstrated. I would suggest its publication on Nature communication after addressing the following comments and issues.

Major issue:

1. The schematic illustration in Figure 1a does not show the enhanced accumulation process after vadimezan treatment which is a very critical.
2. The lung metastatic lesions in Figure 3e are not clear, this reviewer suggest the authors to label them.
3. Please carefully check the quantitative data in Figure 4c&d for Figure 4b. This reviewer noticed that CD4+ positive T cell decreased in G5 treated group Figure 4b.
4. Quantitative results should be given out in Figure 3f.
5. Another animal model may be applied to further support the main conclusions.

Minor issue:

1. Please check which one is right "Blab/c mice" or "BLAB/c" or "BALB/c mice" in 4T1 animal models you used.

Reviewer #3 (Remarks to the Author):

In this manuscript, Li and co-workers demonstrated that directing an anti-programmed cell death-ligand 1 (aPDL1) antibody to assemble in metastatic lesions to promote anti-tumor immune responses when combining with Vadimezan, a tumor blood vessels disrupting molecules in two metastasis models of cancers. The authors were able to generate aPDL1-decorated platelet-derived microparticles (PMP) that diffuse within the tumor and elicit immune responses. In addition, they showed that the combination of Vadimezan and PMP increases 10-folds aPDL1 antibody accumulation in lung metastases as compared to the intravenous administration of the antibody and enhances the magnitude of immune responses

leading to improved anti-tumor effects.

Comments:

Although the work is interesting however there are some issues of this work including, a) the novelty and b) questionable translational impact. The development of platelets decorated with anti-PD-1 antibodies have been described previously by several groups including this group. There are several work showed the implication of this strategy both in solid tumors as well as hematological

malignancies. Hence, this part has been well documented. Secondly, vadimezan had been studied in decades and been tested from preclinical to several clinical trials in combination with chemotherapies, however close to a decade ago, the phase III trial to use as a first line therapy for NSCLC gave poor results and another phase III trial as second-line therapy for NSCLC also gave poor interim results. Importantly, with all those trials, the concept didn't move forward for advancement in clinics. Hence, overall enthusiasm of the significance of the manuscript diminished substantially. Moreover, it is not clear from this work whether the major issue of anti-PD-1 antibody therapy resistance of the cancer patients been addressed.

Other concerns:

a) Interestingly an important control was missing in the experiments related to aPDL1-decorated platelet-derived microparticles (PMP). The authors didn't show any IgG control or other non-specific antibody conjugated with platelets, which would be an important control of the hypothesis.

b) Fig 2, heading of the legend was not clear.

c) In Fig 3, it is not clear regarding the dosing of the different combinations. Did the treatments were done every days for 10 days, then what were the sequences of treatment and what was the original rationale of that treatment.

d) How the PMP was calculated and how much the anti-PD1 antibody was there at the time of therapy.

e) Fig 3F, some figs are out of focus.

f) Fig 5, there was a clear confusion related to designing of the experiments. Interestingly as per Fig 5B, in day 11, just after therapy started for next 10 day, (although the Fig 5A mentioned that for 20 days), we could see some spots in lungs mets in PBS control but other treatment group they very minimum or in G5 group there is no signal. In this regard, the whole experiments presented raised a serious question regarding the study design and rigor of the experiments.

Reviewer #1 (Remarks to the Author):

This well written manuscript describes murine testing of a novel therapy in which anti-PD-L1 is conjugated to platelets. These cell conjugates can deliver more PD-L1 to tumor compared to standard soluble administration when administered with anti-angiogenic compound Vadimezan. The therapeutic was tested in two anti-PD-L1 resistant animal models and demonstrates improved tumor control relative to either the cell conjugate or Vadimezan alone. While the manuscript is well written grammatically, there are a few key issues with the manuscript that greatly weaken it.

1. Importantly, the immunology is very weak overall. There is no evidence that the immune responses observed following administration of the treatment are specific for the tumor (i.e., no antigen-specificity).

Response: We appreciate the reviewer's suggestions. We have added the analysis of immune responses within the tumour. Both the number of CD8⁺IFN- γ ⁺ cytotoxic T cells and the levels of IFN- γ and TNF- α increased in the tumour in mice treated with combinational therapy Vadimezan and Platelet@aPDL1 compared with control groups (Figure R1). These new data suggest that the immune response is also enhanced within the tumor.

In addition, we assessed T cell responses after the administration of Vadimezan and Platelet@aPDL1 in non-tumour bearing mice. No increases of CD3, CD4, CD8 T cells were observed compared with control group further indicating that immune responses observed following the administration of the treatment are tumour specific (Figure R2).

Figure R1. Representative flow cytometry analysis of CD8⁺IFN- γ ⁺ cytotoxic T cells within the tumour in the B16F10 melanoma metastatic model (a) and 4T1 breast metastatic model (b). Detection of IFN- γ (c) and TNF- α (d) within the 4T1 breast metastatic model after treatment. G1, PBS; G2, aPDL1; G3, Vadimezan; G4, P@aPDL1; G5, P@aPDL1 + Vadimezan, G6, P@IgG + Vadimezan (n=5). Data are presented as mean \pm SD, statistical significance was analyzed *via* ANOVA (one-way, Tukey post-hoc test). *P* value: * *P* < 0.05, ** *P* < 0.01, *** *P* < 0.001. (a) and (b) were added as Figure S10 and Figure S15 in the revised version, respectively.

Figure R2. Representative flow cytometry analysis of CD3⁺, CD4⁺, and CD8⁺ T cells in mice treated with PBS or Vadimezan + platelet@aPDL1 (Treated). These data were added as Figure S12 in the revised version.

- Some of the immunologic data do not make sense. For example, the authors claim that a single treatment results in nearly 40% of the circulating CD8 T cells as becoming activated by IFN gamma production. There are no other assay results that support this finding.

Response: In addition to the quantification of IFN- γ expressing CD8 T cells, we also measured the IFN- γ level in the tumour and peripheral blood (Fig. 4h. Fig. S11). Results showed that the combined treatment significantly increased the IFN- γ levels in the tumour and peripheral blood.

- In another example, in Figure 5F, the authors claim that CD3 T cells increase in the peripheral blood by ~50 with the combination. However, the scattergrams are not typical showing a homogeneous group of cells by side scatter analysis. This is not usual pattern seen in blood of mice.

Response: We have optimized the gating strategy to make a more accurate analysis (Figure R3).

Figure R3. (a) Optimized flow cytometry gating strategy. (b) Revised CD3 T cell flow cytometry analysis (Figure 5F). G1, PBS; G2, aPDL1; G3, Vadimezan; G4, P@aPDL1; G5, P@aPDL1 + Vadimezan, G6, P@IgG + Vadimezan.

- Furthermore, the changes are reported as fractional changes which may suggest that there was either a lymphoproliferation or there was a decrease in some other population. Thus, the nature of these changes is unclear.

Response: We agree with the reviewer that the observed changes are likely multifactorial. The PD1/PDL1 pathway is disrupted by the aPDL1 antibody and relieves T cell inhibition¹. T cells reactivated after PD1/PDL1 blockade can in turn secrete cytokines such as interleukin (IL)-2 that promotes T cell proliferation, while IFN- γ can enhance the activity of T helper-1 cells and the adaptive immune response²⁻⁴.

- It is also unclear if any of the changes observed are linked to tumor as the experimental design lacked important controls including platelets conjugated to an irrelevant antibody and testing of the compound in non-tumor bearing animals.

Response: We have added a control group: Vadimezan + platelet@IgG. As shown in Figure 3, Figure 4, and Figure 5 in the revised manuscript, the treatment of Vadimezan + platelet@IgG does not induce significant benefits compared to other control groups.

We have also evaluated the immune response in healthy non-tumour bearing mice treated with of the Vadimezan + platelet@aPDL1. As shown in Figure R2, there are no significant differences in the number of CD3⁺, CD4⁺, and CD8⁺ T cells between the Vadimezan + platelet@aPDL1 treated mice and the PBS treated mice.

- On a minor note, The IFN and TNF graphs in S10 look identical and should be checked.

Response: We corrected the typos in the revised manuscript (Figure S11).

Reviewer #2 (Remarks to the Author):

In this manuscript, the authors developed a strategy that guided an anti-programmed cell death-ligand 1 (aPDL1) antibody to efficiently accumulate in metastatic lesions to promote anti-tumor immune responses. Briefly, by giving a vasculature disruption agent to induce local hemorrhage, activation of the coagulation cascade in advance, the subsequent administrated aPDL1-conjugated platelets (P@aPDL1) can significantly target to metastasis sites. Overall, this is an interesting and novel piece of work and the major conclusions are fully supported and demonstrated. I would suggest its publication on Nature communication after addressing the following comments and issues.

Major issue:

1. The schematic illustration in Figure 1a does not show the enhanced accumulation process after vadimezan treatment which is a very critical.

Response: We revised the schematic illustration to show the platelet accumulation process.

2. The lung metastatic lesions in Figure 3e are not clear, this reviewer suggest the authors to label them.

Response: We labeled the tumour regions with dotted lines in the revised version.

3. Please carefully check the quantitative data in Figure 4c&d for Figure 4b. This reviewer noticed that CD4+ positive T cell decreased in G5 treated group Figure 4b.

Response: Figure 4c illustrates the percentage of CD4+ cells in all lymphocytes (CD45+ cells); figure 4b shows CD4+ cells in CD3+ T cells. Because the number of CD3+ T cells in G5 is higher than in other groups, even if CD4+ cells decrease in CD3+ T cells, the number of CD4+ T cells in lymphocytes remains the highest value compared to other groups.

4. Quantitative results should be given out in Figure 3f.

Response: We have added the quantitative analysis in the revised Figure R4/Figure 3e.

Figure R4. Metastasis index in the lung after treatment. G1, PBS; G2, aPDL1; G3, Vadimezan; G4, P@aPDL1; G5, P@aPDL1 + Vadimezan, G6, P@IgG + Vadimezan. n=5. Data are presented as mean ±

SD, statistical significance was analyzed *via* ANOVA (one-way, Tukey post-hoc test). *P* value: ** $P < 0.01$, *** $P < 0.001$. This data was added as Figure 3e in the revised version.

5. Another animal model may be applied to further support the main conclusions.

Response: We added the Lewis lung metastasis tumour model to further indicate the broad applicability of the proposed strategy (Fig. R5).

Figure R5. Survival of mice bearing Lewis lung cancer metastasis receiving treatments. G1, PBS; G2, aPDL1; G3, Vadimezan; G4, P@aPDL1; G5, P@aPDL1 + Vadimezan, G6, P@IgG + Vadimezan. This data was added as Figure S16 in the revision.

Minor issue:

1. Please check which one is right “Blab/c mice” or “BLAB/c” or “BALB/c mice” in 4T1 animal models you used.

Response: We checked the manuscript thoroughly and corrected these errors.

Reviewer #3 (Remarks to the Author):

In this manuscript, Li and co-workers demonstrated that directing an anti-programmed cell death-ligand 1 (aPDL1) antibody to assemble in metastatic lesions to promote anti-tumor immune responses when combining with Vadimezan, a tumor blood vessels disrupting molecules in two metastasis models of cancers. The authors were able to generate aPDL1-decorated platelet-derived microparticles (PMP) that diffuse within the tumor and elicit immune responses. In addition, they showed that the combination of Vadimezan and PMP increases 10-folds aPDL1 antibody accumulation in lung metastases as compared to the intravenous administration of the antibody and enhances the magnitude of immune responses leading to improved anti-tumor effects.

Comments:

Although the work is interesting however there are some issues of this work including, a) the novelty and b) questionable translational impact. The development of platelets decorated with anti-PD-1 antibodies have been described previously by several groups including this group. There are several work showed the implication of this strategy both in solid tumors as well as hematological malignancies. Hence, this part has been well documented. Secondly, vadimezan had been studied in decades and been tested from preclinical to several clinical trials in combination with chemotherapies, however close to a decade ago, the phase III trial to use as a first line therapy for NSCLC gave poor results and another phase III trial as second-line therapy for NSCLC also gave poor interim results. Importantly, with all those trials, the concept didn't move forward for advancement in clinics. Hence, overall enthusiasm of the significance of the manuscript diminished substantially. Moreover, it is not clear from this work whether the major issue of anti-PD-1 antibody therapy resistance of the cancer patients been addressed.

Response: Thanks so much for the reviewer's valuable comments. Our previous study showed that immune checkpoint blockade anti-PDL1 loaded platelets can be recruited to the wound site after surgical removal of the primary tumour, thereafter platelets were activated and generated platelet-derived microparticles accompanied with anti-PDL1 release to enhance the T cell functions⁵. However, applying this technology in metastatic models remains challenging.

In this study, we leveraged the effect of Vadimezan in disrupting tumor blood vasculature promoting local bleeding, which allows the recruitment of the anti-PDL1 engineered platelets. Our strategy therefore provides a promising active metastasis-targeting drug delivery solution.

The clinical trials with Vadimezan combined with chemotherapies did not show clinical benefits. In contrast, our combination link the activity of Vadimezan in causing tumour local bleeding with the physiologic properties of platelets to participate in blocking the bleeding. We expected this approach could provide clinical benefits. Indeed, we observed 10-folds aPDL1 antibody accumulation in lung metastases. We acknowledge that our propose strategy does not specifically address the issues of aPDL1 resistance.

Other concerns:

a) Interestingly an important control was missing in the experiments related to aPDL1-decorated platelet-derived microparticles (PMP). The authors didn't show any IgG control or other non-specific antibody conjugated with platelets, which would be an important control of the hypothesis.

Response: We have added a control group: Vadimezan + platelet@IgG. As shown in Figure 3, Figure 4, and Figure 5 in the revised manuscript, the treatment of Vadimezan + platelet@IgG does not induce significant benefits compared to other control groups.

b) Fig 2, heading of the legend was not clear.

Response: We have revised the legend's as "P@aPDL1 recruitment at the tumour site after Vadimezan treatment".

c) In Fig 3, it is not clear regarding the dosing of the different combinations. Did the treatments were done every days for 10 days, then what were the sequences of treatment and what was the original rationale of that treatment.

Response: In this study, all mice received one single treatment. Mice were treated with the therapeutics at day 10 because we found that the metastasis formed after 8 days tumour cell inoculation. We have added the treatment dosage in the figure legend: mice received 1 mg/kg aPDL1 antibody or 15 mg/kg Vadimezan.

d) How the PMP was calculated and how much the anti-PD1 antibody was there at the time of therapy.

Response: We used ELISA to measure the dose of aPDL1 on the platelets. The treatment dose is 1 mg aPDL1 per kg mouse body weight. Unfortunately, our current techniques cannot calculate the number of PMP due to their small size (0.1–1 μm).

e) Fig 3F, some figs are out of focus.

Response: We have re-acquired these figures (Figure 3g).

f) Fig 5, there was a clear confusion related to designing of the experiments. Interestingly as per Fig 5B, in day 11, just after therapy started for next 10 day, (although the Fig 5A mentioned that for 20 days), we could see some spots in lungs mets in PBS control but other treatment group they very minimum or in G5 group there is no signal. In this regard, the whole experiments presented raised a serious question regarding the study design and rigor of the experiments.

Response: We have optimized the scale range of the color bar for the bioluminescence image (from $10^5 - 2 \times 10^7$ to $2 \times 10^4 - 2 \times 10^7$), to show the tumour signals. However, tumour signals exceed the body range of the mice due to the low minimum threshold, and for this reason we set the threshold as 10^5 as previously indicate in the figure. Overall, we agree that the current scale range ($2 \times 10^4 - 2 \times 10^7$) could be more accurate. We have measured the total bioluminescence intensity in the lung area at day 11 and the result demonstrated that there is no significant difference in these groups (Figure R6).

Figure R6. (a) In vivo tumour bioluminescence images of mice receiving different treatments. (b) Quantitative analysis of the tumour bioluminescence intensity at day 11, n=5.

Reference:

- 1 Darwin, P., Toor, S. M., Sasidharan Nair, V. & Elkord, E. Immune checkpoint inhibitors: recent progress and potential biomarkers. *Exp Mol Med* **50**, 1-11, doi:10.1038/s12276-018-0191-1 (2018).
- 2 Bhat, P., Leggatt, G., Waterhouse, N. & Frazer, I. H. Interferon-gamma derived from cytotoxic lymphocytes directly enhances their motility and cytotoxicity. *Cell Death Dis* **8**, e2836, doi:10.1038/cddis.2017.67 (2017).
- 3 Dienz, O. & Rincon, M. The effects of IL-6 on CD4 T cell responses. *Clin Immunol* **130**, 27-33, doi:10.1016/j.clim.2008.08.018 (2009).
- 4 Ross, S. H. & Cantrell, D. A. Signaling and Function of Interleukin-2 in T Lymphocytes. *Annu Rev Immunol* **36**, 411-433, doi:10.1146/annurev-immunol-042617-053352 (2018).
- 5 Wang, C. *et al.* In situ activation of platelets with checkpoint inhibitors for post-surgical cancer immunotherapy. *Nature Biomedical Engineering* **1**, doi:10.1038/s41551-016-0011 (2017).

REVIEWERS' COMMENTS

Reviewer #1 (Remarks to the Author):

All of my concerns have been adequately addressed.

Reviewer #2 (Remarks to the Author):

The authors have addressed most issues raised by the reviewers. The reviewer feels satisfactory for this revision.

Reviewer #3 (Remarks to the Author):

The authors tried to respond several concerns raised by the reviewers. However some issues remained questionable.

a) The novelty question of the whole study is still not clarified properly. Give an example, the total improvement of effector CD8T+ cells infiltration have been shown from 4% from the controls to the 5% with the treatment group (e.g. Fig 5i). Although the Fig showed significance however in reality just overall 1% increase in overall how it will really impact in the hypothesis.

b) Another big issue was unaddressed that vascular disrupting agent caused the tumor hypoxic and might cause more problematic. That was one of the problem in the failure of Phase III trial. So, that might be a real issue in the overall hypothesis.

Reviewer #1 (Remarks to the Author):

All of my concerns have been adequately addressed.

Response: We appreciate the reviewer's valuable comments for the improvement of our manuscript.

Reviewer #2 (Remarks to the Author):

The authors have addressed most issues raised by the reviewers. The reviewer feels satisfactory for this revision.

Response: We appreciate the reviewer's valuable comments for the improvement of our manuscript.

Reviewer #3 (Remarks to the Author):

The authors tried to respond several concerns raised by the reviewers. However some issues remained questionable.

a) The novelty question of the whole study is still not clarified properly. Give an example, the total improvement of effector CD8T+ cells infiltration have been shown from 4% from the controls to the 5% with the treatment group (e.g. Fig 5i). Although the Fig showed significance however in reality just overall 1% increase in overall how it will really impact in the hypothesis.

Response: The Fig. 5i shows the ratio of effector memory CD8⁺ T cells is calculated based on CD8⁺ T cells, however, the number of effector memory CD8⁺ T cells should be "Total lymphocytes × %(CD3⁺ T cells) × %(CD8⁺ T cells) × %(CD62L⁻CD44⁺ T cells)". Thus, the number of effector memory CD8⁺ T cells in the Vadimezan + P@aPDL1 group is three times than that in the other control groups.

b) Another big issue was unaddressed that vascular disrupting agent caused the tumor hypoxic and might cause more problematic. That was one of the problem in the failure of Phase III trial. So, that might be a real issue in the overall hypothesis.

Response: Thanks for the reviewer's comment, certain articles and reviews have discussed the reason for Vadimezan failure in Phase III trial, a critical issue was that Vadimezan failure to bind and activate human STING receptor induce its failure in clinical trials(1, 2). In this study, we utilized the Vadimezan to disrupt the tumor vasculature to promote the aPDL1 engineered platelet recruitment and activation for aPDL1 targeting delivery. Our results have demonstrated the feasibility of the concept, and the mechanism of our strategy is STING independent. But more factors, such as tumor hypoxia induced by Vadimezan will be considered in our further studies.

1. J. Conlon, D. L. Burdette, S. Sharma, N. Bhat, M. Thompson, Z. Jiang, V. A. Rathinam, B. Monks, T. Jin, T. S. J. T. J. o. I. Xiao, Mouse, but not human STING, binds and signals in response to the vascular disrupting agent 5, 6-dimethylxanthenone-4-acetic acid. *The Journal of Immunology* **190**, 5216-5225 (2013).
2. A. Daei Farshchi Adli, R. Jahanban-Esfahlan, K. Seidi, S. Samandari-Rad, N. J. C. b. Zarghami, d. design, An overview on Vadimezan (DMXAA): The vascular disrupting agent. *Chemical Biology & Drug Design* **91**, 996-1006 (2018).